# A theoretical model of neural maturation in the developing chick spinal cord

**Piyush Joshi**[1], **Isaac Skromne**[2]*

**1** Division of Pediatric Neuro-oncology, German Cancer Research Center (DKFZ), Heidelberg, Germany,
**2** Department of Biology, University of Richmond, Richmond, Virginia, United States of America

* iskromne@richmond.edu

**Data Availability Statement:** All relevant data are within the manuscript and its Supporting Information files.

**Funding:** P.J. was supported by Sigma XI GIAR, and I.S. was supported by University of Richmond School of Arts and Sciences and by the National

## Abstract

Cellular differentiation is a tightly regulated process under the control of intricate signaling and transcription factors interaction network working in coordination. These interactions make the systems dynamic, robust and stable but also difficult to dissect. In the spinal cord, recent work has shown that a network of FGF, WNT and Retinoic Acid (RA) signaling factors regulate neural maturation by directing the activity of a transcription factor network that contains CDX at its core. Here we have used partial and ordinary (Hill) differential equation based models to understand the spatiotemporal dynamics of the FGF/WNT/RA and the CDX/transcription factor networks, alone and in combination. We show that in both networks, the strength of interaction among network partners impacts the dynamics, behavior and output of the system. In the signaling network, interaction strength determine the position and size of discrete regions of cell differentiation and small changes in the strength of the interactions among networking partners can result in a signal overriding, balancing or oscillating with another signal. We also show that the spatiotemporal information generated by the signaling network can be conveyed to the CDX/transcription network to produces a transition zone that separates regions of high cell potency from regions of cell differentiation, in agreement with most *in vivo* observations. Importantly, one emerging property of the networks is their robustness to extrinsic disturbances, which allows the system to retain or canalize NP cells in developmental trajectories. This analysis provides a model for the interaction conditions underlying spinal cord cell maturation during embryonic axial elongation.

## Introduction

Cells sequentially differentiate from high to low potency states, under the guidance of extracellular signals working in coordination with intracellular transcription factors. Signals regulate the individual and network activity of the transcription factors by providing spatial and temporal information [1–4]. In turn, transcriptional network dictates a cell's competence and response to extracellular signals [5–7]. Because signaling information changes the composition of a cell's transcriptional components, this creates an intricate and dynamic cross-regulatory system for guiding cell differentiation that has been challenging to untangle and comprehend [1, 3, 4]. Understanding the cross-regulatory interactions between signal and transcription

Science Foundation (IOS-1755386). The funders had no role in study design, data collection and analysis, decision to publish, or preparation of the manuscript.

factor sub-networks will be important for understanding to how cell trajectories are retained during development in the face of genetic or environmental perturbations (canalization; [8])

Vertebrate spinal cord provides an advantageous model to study the cross-regulatory dynamics involved in central nervous system development in particular, and differentiation in general. The head (rostral) to tail (caudal) development of spinal cord during vertebrate body extension results into a characteristic spatial separation of temporal differentiation events [9–11], facilitating the study of their regulation. Experimental data obtained from mouse, chick and zebrafish embryos support a model in which spinal cord neural progenitors (NPs) are derived from a bipotent population of cells located at the caudal most end of the embryo, the neuro-mesodermal progenitors (NMPs) cells [9, 10]. In the early embryo, the region where NMPs reside is known as the caudal lateral epiblast and node streak border, and once the tailbud has formed in the late embryo (in chick between 16–22 somite stage; [12, 13]), the caudal neural hinge [9–11]. During development, NP cells exit the NMP domain rostrally and then, sequentially, transit through different maturation states as they become part of the spinal cord [9, 14, 15].

NP cell maturation is driven by synergistic and antagonistic interactions between the signaling factors FGF, WNT and Retinoic Acid (RA), turning on and off key transcription factors required for caudal-to-rostral maturation events (Fig 1A). In the chick trunk region of the spinal cord (somites 6–18), two opposite signaling gradients are proposed to regulate spinal cord cell maturation [16]: from caudal/high to rostral/low, FGF and WNT gradients prevent cell differentiation by promoting high potency cell states caudally; whereas an opposite rostral/high to caudal/low gradient of RA secreted from somites promotes cell differentiation rostrally. Importantly, FGF and WNT activity gradient counteract RA activity gradient. In this way, NMP cells located caudally experience high levels of FGF/WNT and no RA, which drives expression of bipotency markers *T/Bra*, *Sox2*, and *Nkx1.2* (*Sax1*) [16–18]. T/BRA and SOX2 are transcription factors that repress each other and promote different cell fates, with T/BRA promoting mesoderm and SOX2 promoting neural fates [17–19], a phenomenon extensively documented in mouse [20–23]. In addition, both T/BRA and SOX2 can downregulate FGF and WNT pathway activity, initiating the early differentiation of mesoderm or neural tissues [20]. NMPs that continue to transcribe *Sox2* but not *T/Bra* assume NP identity and become part of the growing neural plate. As NPs transit through the maturing neural plate, they experience a further gradual loss in FGF and WNT, and a gradual increase in RA signaling. This new environment lead to the caudal-to-rostral downregulation of a third bipotency marker, *Nkx1.2*, and upregulation of the early differentiation gene *Pax6* [24]. Subsequently, under RA regulation, PAX6 activates late differentiation genes such as *Ngn2* (Fig 1B) [25, 26]. Recently, we have experimentally mapped the interactions between *T/Bra*, *Sox2*, *Nkx1.2*, *Cdx4*, *Pax6* and *Ngn2* into a gene regulatory network (GRN) that we placed it in the context of the FGF/WNT-RA signaling network (Fig 1C) [15]. This work identified the transcription factor CDX4 as a core system component essential for the sequential maturation of NPs into mature neuronal precursors (Fig 1C).

Here we use partial and ordinary (Hill) differential equations to dynamically analyze the GRN driving NP cell maturation during early chick spinal cord development (10–18 somite stage). As the transcription factor network depends upon inputs form the FGF-WNT-RA signaling network, we first analyzed the postulated effectiveness of the signaling network to work as a signaling switch [27]. We then used the resulting signaling dynamics as input to evaluate the performance of the underlying transcription GRN in its ability to generate cell state patterns similar to those observed in experimental models. Our results show that signaling interaction can give rise to various developmentally observed phenotypes based on a limited subset of interaction parameters, and these behaviors are robust and stable to perturbations. Network robustness is a property emerging from strong cross-regulation interactions between

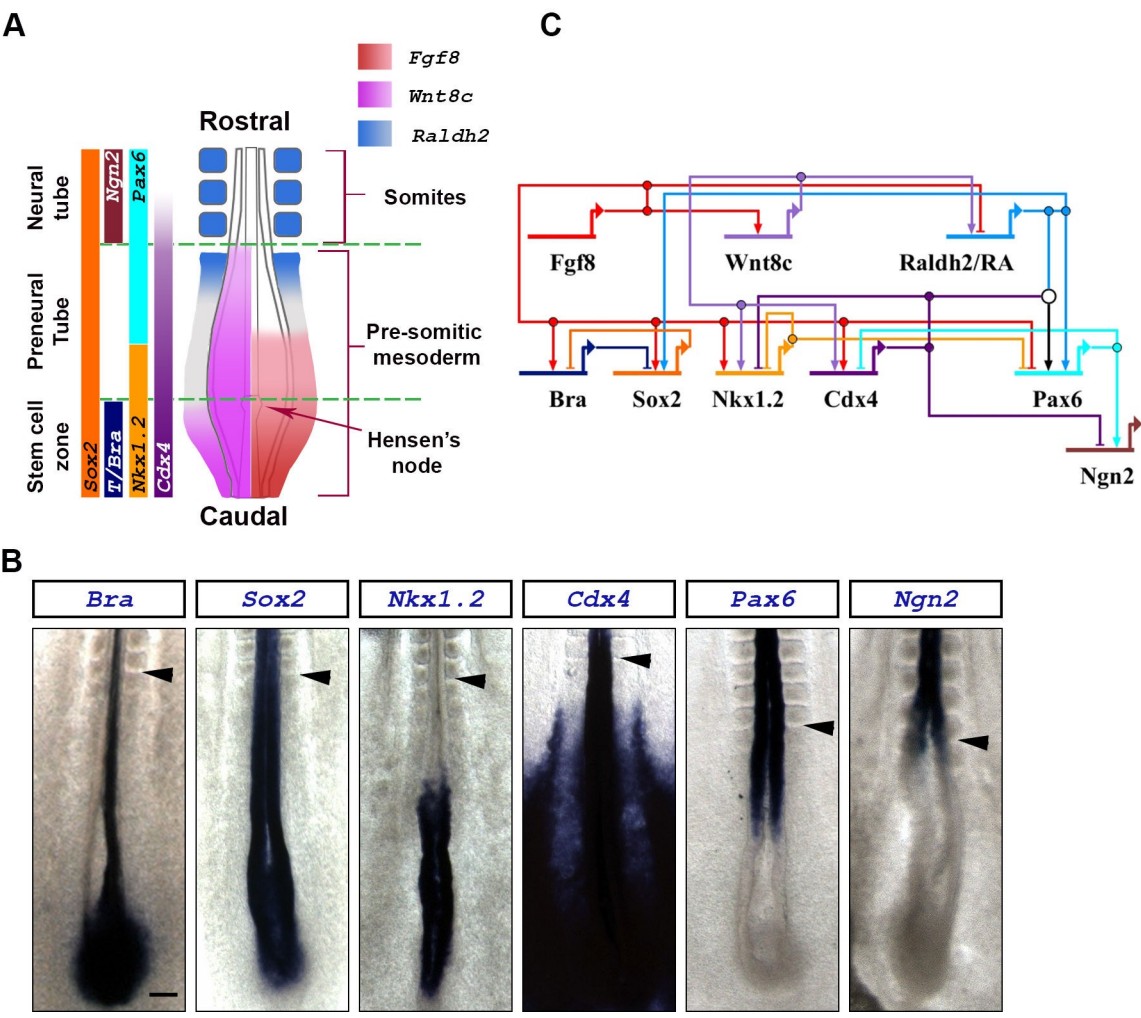

**Fig 1. Expression domains and network interactions of key signaling and transcription factors involved in caudal spinal cord maturation.** (**A**) Schematic representation of the caudal end of a stage HH10-11 chick embryo (dorsal view). Expression domains of *Fgf8* (red) and *Wnt8c* (magenta) signaling factors, and the Retinoic Acid synthesizing enzyme *Raldh2* (blue), are superimposed on the diagram (based on [27]). Expression domain of relevant transcription factors are indicated on the left (based on [15]). (**B**) Expression domains of key transcription factors involved in caudal spinal cord maturation. Embryos are stage HH10-11. Scale bar is 200μm. Arrowheads indicates the anterior boundary of the last formed somite. Transcription of the *T/Bra* gene along the embryo's midline is in the notochord underlying the neural tissue, where it is absent. (**C**) Postulated gene regulatory network showing interaction between signaling and transcription factors (based on [15, 20, 25, 27]).

individual system components whose function we propose is to canalize the cells in their NP trajectories. Our results suggests that the dominant predictor of the GRN response is the interaction strength among network partners. By outlining the conditions that permit the operation of the GRN during NP maturation *in silico*, the model predicts and informs on cellular behaviors of the system *in vivo*.

## Materials and methods

### Chick embryos, gene expression analysis and ethics statement

Fertile broiler chicken eggs (Morris Hatchery, Inc.; Miami, FL) were incubated to the 10-somite stage of development before embryos were processed for expression analysis (about

35 hours at 38˚C in a humid chamber). Expression analysis of relevant genes was done as previously reported (see S1 File; [15]). Chick embryos younger than three days, such as the ones used in this study, are considered by The American Association for Laboratory Animal Sciences (AALAS) and the American Veterinary Medical Association (AVMA), incapable of feeling pain. Therefore, this study is exempt of Institutional Animal Care and Use Committees (IACUC) review.

## Model outline

Our model aims to describe the maturation of NP cells in the pre-neural tube region of chick embryos between 10–18 somite stage using previously published empirical data [15]. This time period was selected for several reasons. First, the average velocity of axial elongation (2.5–3μm/min) and the size pre-neural tube region (2500 microns) are relatively consistent (measured from the caudal lateral epiblast where the NMPs reside to where the neural tube closes at the anterior boundary of the last formed somite) [28]. Second, most parametric values required by the model are available for the chick embryo, and the few missing ones can be extrapolated from mouse or cell culture data (described in detail below). Third, the only *Cdx* family member transcribed in the chick embryo between 10–18 somite stage is *Cdx4* [29]. Finally, we could overlook GDF11 activity in terminating axial elongation, as this activity in mouse is associated with the relocation of NMP from the caudal primitive streak epiblast to the tail bud [30, 31], which in chicks occurs after the stages our simulation models (16–22 somite stage; [13]). In addition, our model assumes that NP production to occur at a steady rate, independently of any network components. This is not the case *in vivo*, were experimental evidence suggests an involvement of NOTCH signaling pathway in this process [32, 33]. This assumption was made due to paucity of evidence connecting NOTCH regulation to *Cdx4* and many of the transcription factors in the GRN.

## Hill equation based interaction model

Signal and transcription factor networks were modeled using ordinary and partial differential Hill equations. Ordinary differential Hill equations were used to model molecules whose rate of change is not influenced by diffusion (e.g., mRNA and intracellular proteins), and partial differential Hill equations for molecules whose rate of change in a field is contingent on their diffusion (e.g., extracellular factors) [34–36]. We first modeled the signal interactions network, using the resulting output as the input for the transcription factor network. To solve the equations numerically and plot the simulations we used MATLAB (MathWorks, Natick, MA) with solvers *ode45* for ordinary and *pdepe* for partial differential equations. Within each Hill equation, a number of Hill constants were used to vary the strength of interaction between a molecule and its target (e. g., transcription factor and its target gene; S1 Fig). These equations follow the general form;

Ordinary differential equation to model mRNA dynamics:

$$\frac{\partial M}{\partial t} = \alpha_m H_1 - \beta_m M H_2$$

Ordinary differential equation to model intracellular protein dynamics (e.g., transcription factors):

$$\frac{\partial P}{\partial t} = \alpha_p M H_3 - \beta_p P H_4$$

Partial differential equations to model dynamics of extracellular factors (e.g., signaling molecules):

$$\frac{\partial P}{\partial t} = \alpha_p M H_3 - \beta_p P H_4 - \mu \frac{\partial P}{\partial x^2}$$

where,

$\alpha_m$ = Transcription rate constant $\alpha_p$ = Translation rate constant

$\beta_m$ = mRNA decay rate constant $\beta_p$ = protein decay rate constant

$M$ = mRNA concentration $P$ = protein concentration

$\mu$ = diffusivity coefficient $x$ = spatial dimension

$H_1$, $H_2$, $H_3$, $H_4$ are independent Hill functions. For each factor being modeled, we replaced $H_1$, $H_2$, $H_3$, and $H_4$ with one of four general types of functions representing regulatory interaction observed *in vivo* (see S1 File): (1) inductive interactions from one or multiple activators, (2) repressive interactions from one or multiple repressors, (3) coordinated interactions between activators and repressors binding to separate regulatory sites and (4) competitive interactions between activators or repressors binding to the same regulatory sites.

## Equations modeling the signaling interactions network

Partial differential Hill equations that take diffusion into consideration were used to model FGF8, WNT8C and RA network of interactions (Fig 2A). These interactions were modeled within a spatial maturation domain restricted to a 2500 microns extending from the NMP zone to the anterior boundary of the last formed somite (stage HH10-11 embryos; Figs 1B and 2B). This spatial maturation domain moves caudally and in synchrony with the NMP zone during axial elongation (constant velocity), thus appearing stationary with respect to the NMP zone (Fig 2B). When available, we used parameter values that have been determined experimentally, within reported ranges. For parameters that have not been determined experimentally (e.g., rates constants for mRNA and protein synthesis and degradation), we used parameters values comparable to those used in other models [28, 37, 38]. We set the Hill coefficients value for FGF and Wnt at 2, as empirically established in somitogenesis network model [37]. For RA, the value of the Hill coefficient factor was set at 2, as RA's receptor is a transcription factor that operates as a dimer [39].

**FGF8 production.** *Fgf8* transcription is restricted to the NMP zone through positive autoregulatory loops and inhibitory signals [40]. FGF8 indirectly stimulates its own transcription by inducing transcription of *Nkx1.2*, *Cdx*, and WNT/ß-catenin pathway components [40]. RA secreted from somites restricts *Fgf8* to the NMP zone in a concentration-dependent manner [16]. RA inhibition is excluded from NMP zone by CYP26A, an RA-catabolizing enzyme whose gene is activated by FGF8 [18]. We simulated *Fgf8* positive autoregulatory loop by assuming a basal exponential level of gene transcription, and its restriction to the caudal end of the spatial maturation domain by allowing RA to decrease *Fgf8* transcription down to zero in a concentration-dependent manner.

*Fgf8* mRNA transcripts have a long half-life of around 2 hours, persisting in cells long after transcription has stopped [37, 41]. This long decay results in a graded distribution of transcript in the spatial maturation domain, with cells proximal to the NMP zone retaining more transcripts that more distal cells. This 2 hour half-life sets the rate constant of degradation to around 0.006 min$^{-1}$ (ln2/2h = 0.693/120min). As the average speed of axis elongation is 2.5–3μm/min (from somite 5 to 18; [28]), the decay constant in the spatial maturation domain is about 0.002 μm$^{-1}$.

FGF8 protein synthesis is dependent on the concentration of the *Fgf8* transcript within each cell. As FGF8 is synthesized, it diffuses from producing cells at a rate that has been

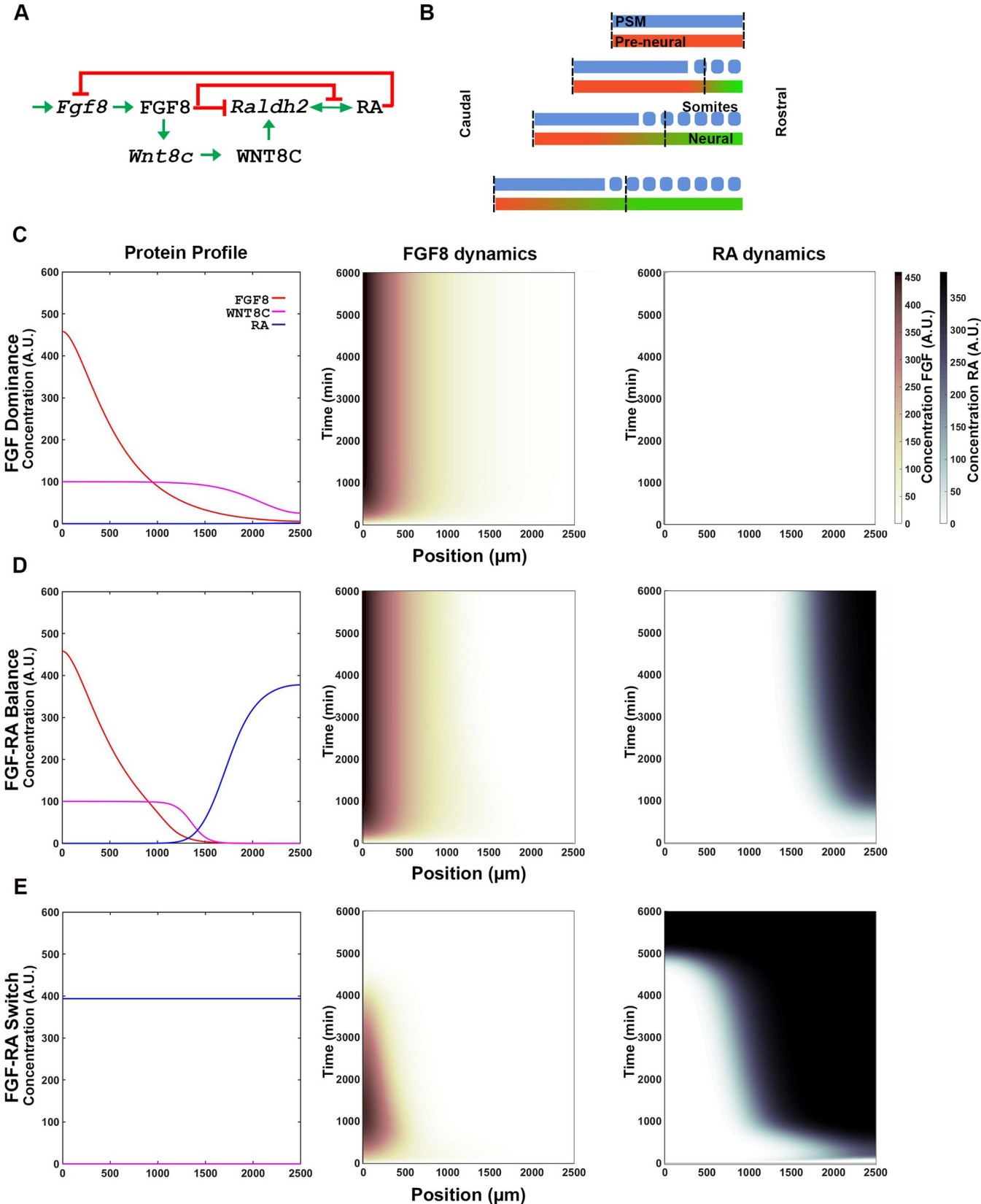

**Fig 2. Signaling network output is determined by the strength of interactions between FGF, WNT, and RA pathway components.** (**A**) FGF8, WNT8C and RA signaling pathway interaction network based on [27]. Names in lower case indicate mRNA and upper case proteins (FGF8 and WNT8C) or metabolites (RA). (**B**) The spatial maturation domain where the signaling network operates extends from the NMP cells to the anterior boundary of the last formed somite (vertical dashed lines; x-axes on graphs). The domain in the simulation has a constant length maintained by a caudal movement that is equivalent to the rate of NMP cell proliferation. Initially undifferentiated cells differentiate at a rate defined by the simulation (red to green transition). (**C-E**) Representative FGF8 dominant (C), FGF-RA balance (D), and FGF-RA switch (E) simulation profiles obtained using parameters shown in Table 1. Left graphs shows the levels of the signaling molecules FGF8 (red), WNT8C (magenta) and RA (blue) across the maturation domain at the end of the simulation (t = 6000 min; arbitrary units AU). Center and right graphs show heat maps of FGF8 (center) and RA (right) accumulation in the maturation domain (x-axis) over time (y-axis). AU scale for FGF (maroon gradient) and RA (blue gradient) are shown at right of graphs.

determined experimentally to be around 2 μm$^2$/sec [42]. Due to this diffusion, the domain of FGF protein signaling expands beyond the domain of *Fgf8* transcription.

Constant input: $F_0(x)$

$$F_0 = 0.06e^{-.002x} \tag{1}$$

*Fgf8* mRNA transcription: $F_m(t)$

$$\frac{\partial F_m}{\partial t} = \alpha_{Fm}F_0\left(\frac{1}{(1 + (R/R_{FR})^r)}\right) - \beta_{Fm}F_m \tag{2}$$

FGF8 translation: $F(x, t)$

$$\frac{\partial F}{\partial t} = \alpha_{Fp}F_m - \beta_{Fp}F - D_F\frac{\partial^2 F}{\partial x^2} \tag{3}$$

where,

*Fgf8* mRNA transcription rate constant [37, 38] $\alpha_{Fm}$ = 1/min
*Fgf8* mRNA half-life [37] $\beta_{Fm}$ = 0.006/min.
FGF8 translation rate constant [37] $\alpha_{Fp}$ = 0.3/min
FGF8 degradation rate constant [37] $\beta_{Fp}$ = 0.005/min
FGF8 diffusion constant [42] $D_F$ = 120 μm$^2$/min
Hill constant, *Fgf8* inhibition by RA $R_{RF}$ (see Table 1)

**WNT8C production.** *Wnt8c* transcription is stimulated by FGF pathway activity and is indirectly blocked by RA inhibiting *Fgf8* transcription [27]. In chick embryos, *Wnt8c*

**Table 1. Examples of Hill constants combinations tested to investigate signaling dynamics behavior.**

| Hill constants | I | II | III | IV | V | VI | VII | VIII |
|---|---|---|---|---|---|---|---|---|
| $F_{FW}$ | 10 | 10 | 10 | 10 | 10 | 10 | 10 | 10 |
| FGF dependent activation of *Wnt8c* transcription | | | | | | | | |
| $F_{FR1}$ | 1 | 1 | 5 | 10 | 10 | 10 | 2 | 20 |
| FGF dependent repression of *Raldh2* transcription | | | | | | | | |
| $F_{FR2}$ | 2 | 10 | 15 | 10 | 10 | 10 | 20 | 20 |
| FGF dependent activation of RA degradation (via CYP26A enzymes) | | | | | | | | |
| $W_{WR}$ | 1 | 1 | 0.2 | 0.5 | 0.2 | 0.5 | 1 | 1 |
| WNT dependent activation of *Raldh2* transcription | | | | | | | | |
| $R_{RF}$ | 10 | 1 | 0.2 | 0.1 | 0.3 | 0.45 | 1 | 20 |
| RA dependent repression of *Fgf8* transcription | | | | | | | | |
| $R_{RR}$ | 50 | 50 | 50 | 50 | 50 | 50 | 300 | 300 |
| RA dependent activation of *Raldh2* transcription | | | | | | | | |
| Outcome (t = 6000 min) | FGF dominant | FGF-RA balance | FGF-RA switch | | | | RA aberrant/ oscillatory | |

expression domain significantly overlaps and extends far beyond *Fgf8* mRNA domain up to the last formed somite (Fig 1A; [27, 43]), suggesting that very low levels of FGF8 can activate *Wnt8c* [27, 43]. By contrast, in the NMP zone, *Wnt8c* mRNA level are one-third of the *Fgf8* mRNA level [44], which suggest that transcription rate constant of *Wnt8c* are low and saturate quickly [44]. Once synthesized, WNT8C diffuse from its site of synthesis throughout the spatial maturation domain at a low diffusion rate [45].

*Wnt8c* mRNA transcription: $W_m(t)$

$$\frac{\partial W_m}{\partial t} = \alpha_{Wm}\left(\frac{(F/F_{FW})^a}{(1+(F/F_{FW})^a)}\right) - \beta_{Wm}W_m \tag{4}$$

WNT8C translation: $W(x, t)$

$$\frac{\partial W}{\partial t} = \alpha_{Wp}W_m - \beta_{Wp}W - D_W\frac{\partial^2 W}{\partial x^2} \tag{5}$$

, where,

  *Wnt8c* mRNA transcription rate constant [27, 43] $\alpha_{Wm} = 0.1$/min
  *Wnt8c* mRNA half-life constant [27, 43] $\beta_{Wm} = 0.03$/min
  WNT8C translation rates constant [37] $\alpha_{Wp} = 0.3$/min
  WNT8C degradation rates constant [37] $\beta_{Wp} = 0.01$/min
  WNT8C diffusion rate rates [45] $D_W = 10$ μm$^2$/min
  Hill constant, *Wnt8c* activation by FGF8 $F_{FW}$ (see Table 1)

**RA production.** RA is synthesized in somites by the enzyme RALDH2 [46]. *Raldh2* transcription is restricted to somites as this is the only region where activation by the WNT8C pathway can overcome FGF8-dependent repression [27]. Parameters for RALDH2 production and degradation were equivalent to those in other models [47]. We assumed that once RALDH2 is produced, RA synthesis initiates without delay. Once produced, RA diffuses into undifferentiated neural and mesodermal tissues at an estimated rate of 18 μm$^2$/sec or about 1080 μm$^2$/min [47]. At the caudal end of the embryo, RA is degraded by the enzyme CYP26A, whose transcription is under FGF8 regulation [25].

*Raldh2* mRNA transcription: $R_m(t)$

$$\frac{\partial R_m}{\partial t} = \alpha_{Rm}\left(\frac{(W/W_{WR})^a + (R/R_{RR})^a}{(1+(W/W_{WR})^a + (R/R_{RR})^a)}\right)\left(\frac{1}{(1+(F/F_{FR1})^r)}\right) - \beta_{Rm}R_m \tag{6}$$

RA production (as modeled by RALDH2 translation): $R(x,t)$

$$\frac{\partial R}{\partial t} = \alpha_{Rp}R_m - \beta_{Rp}R\left(1 + \beta_{FR}\frac{(F/F_{FR2})^a}{(1+(F/F_{FR2})^a)}\right) - D_R\frac{\partial^2 R}{\partial x^2} \tag{7}$$

, where,

  *Raldh2* mRNA transcription rate constant [37] $\alpha_{Rm} = 1$/min
  *Raldh2* mRNA half-life constant [37] $\beta_{Rm} = 0.03$/min
  RALDH2 translation rates constant [37] $\alpha_{Rp} = 0.3$/min
  RALDH2 degradation rates constant [37] $\beta_{Rp} = 0.025$/min
  RA estimated diffusion rate $D_R = 1200$ μm$^2$/min
  FGF dependent RALDH2 degradation constant $\beta_{FR} = 6$/min
  Hill constant, *Raldh2* induction by WNT8C $W_{WR}$ (see **Table 1**)
  Hill constant, *Raldh2* induction by RA (autoregulation) $R_{RR}$ (see **Table 1**)
  Hill constant, *Raldh2* repression by FGF8 $F_{FR1}$ (see **Table 1**)
  Hill constant, RA degradation by FGF8-induced CYP26A $F_{FR2}$ (see **Table 1**)

## Equations modeling the transcription factors interactions network

We used differential equations to simulate the transcription factor network (Fig 1C), as transcription factors do not diffuse outside the cell. As inputs, we used FGF8, WNT8C and RA output levels obtained in the signaling simulation (represented in the equations with the letters F, W and R, respectively). For transcription factors binding as dimers, the Hill coefficient was set to a value of 2 (T/BRA, SOX2, CDX4, PAX6 and NGN2; [48–52]), following other model's practices [37]. For simplicity, we also assumed a Hill coefficient value of 2 for transcription factors with no binding information (NKX1.2, X and Y). Network interactions described in the result section are supported by experimental evidence (reviewed in [15]).

*T/Bra* mRNA: $T_m(t)$
(Synthesis activated by FGF8 and inhibited by SOX2.)

$$\frac{\partial T_m}{\partial t} = \alpha_{Tm}\left(\frac{(F/F_{FT})^a}{(1 + (F/F_{FT})^a + (S/S_{ST})^r)}\right) - \beta_{Tm}T_m \tag{8}$$

T/BRA protein: $T(t)$

$$\frac{\partial T}{\partial t} = \alpha_{Tp}T_m - \beta_{Tp}T \tag{9}$$

*Sox2* mRNA: $S_m(t)$
(Synthesis activated by FGF8 and RA and inhibited by T/BRA.)

$$\frac{\partial S_m}{\partial t} = \alpha_{Sm}\left(\frac{(F/F_{FS})^a + (R/R_{RS})^a}{(1 + (F/F_{FS})^a + (R/R_{RS})^a + (T/T_{TS})^r)}\right) - \beta_{Sm}S_m \tag{10}$$

SOX2 protein: $S(t)$

$$\frac{\partial S}{\partial t} = \alpha_{Sp}S_m - \beta_{Sp}S \tag{11}$$

*Nkx1.2* mRNA: $NK_m(t)$
(*Nkx1.2* is activated by WNT8C and inhibited by a CDX4-dependent factor X and by NKX1.2 protein.)

$$\frac{\partial NK_m}{\partial t} = \alpha_{NKm}\left(\frac{(W/W_{WNK})^a}{(1 + (W/W_{WNK})^a + (NK/NK_{NKNK})^r + (X/X_{XNK})^r)}\right) - \beta_{NKm}NK_m \tag{12}$$

NKX1.2 protein: $NK(t)$

$$\frac{dNK}{dt} = \alpha_{NKp}NK_m - \beta_{NKp}NK \tag{13}$$

*Cdx4* mRNA: $C_m(t)$
(*Cdx4* is induced by FGF8 and WNT8C and inhibited by a PAX6-dependent factor Y.)

$$\frac{\partial C_m}{\partial t} = \alpha_{Cm}\left(\frac{(F/F_{FC})^a + (W/W_{WC})^a}{(1 + (F/F_{FC})^a + (W/W_{WC})^a + (Y/Y_{YC})^r)}\right) - \beta_{Cm}C_m \tag{14}$$

CDX4 protein: $C(t)$

$$\frac{\partial C}{\partial t} = \alpha_{Cp}C_m - \beta_{Cp}C \tag{15}$$

Factor *X* mRNA: $X_m(t)$

(Factor *X* is induced CDX4. We have assumed that *X* is inhibited by high FGF8 levels since repression of *Nkx1.2* by CDX4-dependent Factor X is not effective in NMP zone.)

$$\frac{\partial X_m}{\partial t} = \alpha_{Xm}\left(\frac{(C/C_{CX})^a}{\left(1 + (C/C_{CX})^a + (F/F_{FX})^r\right)}\right) - \beta_{Xm}X_m \tag{16}$$

X protein: *X(t)*

$$\frac{\partial X}{\partial t} = \alpha_{Xp}X_m - \beta_{Xp}X \tag{17}$$

*Pax6* mRNA: $P_m(t)$
(*Pax6* is induced by CDX4 and RA working cooperatively, and is inhibited by NKX1.2)

$$\frac{\partial P_m}{\partial t} = \alpha_{Pm}\left(1 + \frac{(C/C_{CP})^a}{\left(1 + (C/C_{CP})^a + (NK/NK_{NKP})^r\right)}\right)\left(\frac{(R/R_{RP})^a}{\left(1 + (R/R_{RP})^a\right)}\right) - \beta_{Pm}P_m \tag{18}$$

PAX6 protein: *P(t)*

$$\frac{\partial P}{\partial t} = \alpha_{Pp}P_m - \beta_{Pp}P \tag{19}$$

Factor *Y* mRNA: $Y_m(t)$
(Synthesis activated by PAX6, and inhibited by FGF8.)

$$\frac{\partial Y_m}{\partial t} = \alpha_{Xm}\left(\frac{(P/P_{PY})^a}{\left(1 + (P/P_{PY})^a + (F/F_{FY})^r\right)}\right) - \beta_{Ym}Y_m \tag{20}$$

Y protein: *Y(t)*

$$\frac{\partial Y}{\partial t} = \alpha_{Yp}Y_m - \beta_{Yp}Y \tag{21}$$

*Ngn2* mRNA: $N_m(t)$
(Synthesis activated by PAX6 and inhibited by factor X.)

$$\frac{\partial N_m}{\partial t} = \alpha_{Nm}\left(\frac{(P/P_{PN})^a}{\left(1 + (P/P_{PN})^a + (X/X_{XN})^r\right)}\right) - \beta_{Nm}N_m \tag{22}$$

Name definition and values for the Hill constants used in the transcription factor network are found in Table 2. For all these transcription factors, the rate constants of mRNA and protein synthesis and degradation have not been determined experimentally. Hence, all the values are kept similar based on values used in published models [37, 38]. The only exception was CDX4, as CDX proteins are known to have increased stability [53].

Constant for mRNA synthesis/degradation: $\alpha_{im} = 1/$ min $\beta_{im} = 0.03/$ min
Constant for protein synthesis/degradation: $\alpha_{ip} = 1/$ min $\beta_{ip} = 0.2/$ min
CDX4 constant for protein synthesis/degradation: $\alpha_{Cp} = 1/$ min $\beta_{Cp} = 0.05/$ min

## Results

### FGF-WNT-RA signaling interaction network can drive signaling switch

In order to model the transcription factor network responsible for spinal cord cell maturation (Fig 1C), we first simulated the signaling dynamics between FGF, WNT and RA driving the system in the chick caudal neural tube [16, 27]. Although several partially redundant FGF and WNT factors are transcribed within and around the caudal neural plate [27, 54], in chick, the

**Table 2. Hill constant for correct spatiotemporal distribution of cellular states.**

| Hill constants | Description | Value* |
|---|---|---|
| $F_{FT}$ | FGF8 dependent activation of $T$ | 10 |
| $F_{FS}$ | FGF8 dependent activation of $Sox2$ | 50 |
| $F_{FC}$ | FGF8 dependent activation of $Cdx4$ | 5 |
| $F_{FX}$ | FGF8 dependent repression of $X$ | 1 |
| $F_{FY}$ | FGF8 dependent repression of $Y$ | 1 |
| $W_{WN}$ | WNT8C dependent activation of $Nkx1.2$ | 10 |
| $W_{WC}$ | WNT8C dependent activation of $Cdx4$ | 10 |
| $S_{ST}$ | SOX2 dependent repression of $T$ | 2 |
| $T_{TS}$ | T dependent repression of $Sox2$ | 20 |
| $N_{NN}$ | NKX1.2 dependent repression of $Nkx1.2$ | 100 |
| $N_{NP}$ | NKX1.2 dependent repression of $Pax6$ | 20 |
| $C_{CX}$ | CDX4 dependent activation of $X$ | 10 |
| $C_{CP}$ | CDX4-RA complex dependent activation of $Pax6$ | 10 |
| $R_{RS}$ | RA dependent activation of $Sox2$ | 1 |
| $R_{RP}$ | RA dependent activation of $Pax6$ | 10 |
| $X_{XN}$ | X dependent repression of $Nkx1.2$ | 1 |
| $X_{XN2}$ | X dependent repression of $Ngn2$ | 1 |
| $P_{PY}$ | PAX6 dependent activation of $Y$ | 5 |
| $P_{PN2}$ | PAX6 dependent activation of $Ngn2$ | 20 |
| $Y_{YC}$ | Y dependent repression of $Cdx4$ | 5 |

*Correct spatiotemporal distribution of cellular states was also obtained when individual values are increased or decreased by 30%.

most relevant factors are FGF8 and WNT8C [27]. *Fgf8* is transcribed in the caudal stem zone (Fig 1A), where it activates *Wnt8c* transcription [27] and represses RA by inhibiting transcription of the RA synthesis enzyme *Raldh2* and by activating transcription of the RA degradation enzyme *Cyp26a* [25]. FGF8 inhibition of RA production is circumvented rostrally by *Fgf8* mRNA decay [41] and by WNT8C, which stimulates RA production by outcompeting FGF8-mediated *Raldh2* repression [27]. Once *Raldh2* induction has occurred in nascent somites, its expression is maintained through unknown mechanisms, even in the absence of WNT activity [27]. For simplification, our model assumes that RA maintains *Raldh2* transcription through positive autoregulation [55]. RA produced by somites then diffuses caudally and inhibit *Fgf8* transcription [25, 56]. These interactions give rise to an extended negative feedback loop between FGF and RA (Fig 2A).

At the stages examined, cell proliferation in the stem zone extend the vertebrate body axis caudally by producing the cells that, upon maturation, will give rise to the embryo's trunk [9]. To simulate the tissue's caudal ward movement, the signaling interactions were confined to a caudally moving spatial maturation domain of constant length extending rostrally from the stem cell zone to the anterior boundary of the most recently formed somite (Fig 2B; [28]). Thus, from the perspective of the caudal end, the moving spatial maturation domains appears stationary. To simulate the interactions between signaling factors, we used partial differential equations that integrated synthesis, degradation, and diffusion constant through interaction parameters or Hill constants. The Hill constant of a given reaction is defined as the concentration of a factor at which the rate of reaction regulated by the factor is half of the maximum possible rate. Hence, Hill constants are inversely related to the affinity of a factor for its target and can act as a measure of the factor's interaction strength (S1 Fig).

To understand the possible behaviors that could originate from the extended FGF-WNT-RA network, we analyzed the system's output after systematically changing the signaling inputs and the strength of interaction between components (strong Hill constant = 0.1 to weak Hill constant = 100). By varying the interaction strength between FGF, WNT and RA components we obtained various temporal signaling information profiles that we grouped into four broad behaviors: FGF-dominance, FGF-RA balance, FGF-RA switch, and RA aberrant/oscillatory.

**FGF8 dominance.**  In a system where FGF8 repression of *Raldh2* transcription outweighs RA repression of *Fgf8* transcription, the interactions do not result in appreciable RA production (e. g., Table 1-I; Fig 2C, S2A Fig). Such a system would lead to maintenance of pluripotent stem progenitor cells without differentiation.

**FGF8-RA balance.**  FGF8, WNT8C and RA signaling domains balance each other and settle on a stable steady state profile (Table 1-II; Fig 2D, S2A and S2B Fig). Such steady state is achieved when the activating and repressive interactions of the system reach an equilibrium. In these conditions, the regions of FGF8 and RA activities are restricted to domains that maintain the same distance from one. This equilibrium could be broken at the onset of tail bud stages of development (18–21 somite stage) by the activation of signals that terminate axial elongation such as GDF11 [30, 31].

**FGF-RA switch.**  One of the most interesting behavior obtained from the simulation is where the system starts with an *Fgf8* mRNA gradient and ends with RA activity gradient over the entire spatial domain (Table 1-III; Fig 2E, S2A Fig). This behavior simulates a system that starts with a caudally located stem cell zone and a field of undifferentiated cells that is gradually converted, in a rostral to caudal direction, to a field of differentiate cells. Significantly, this differentiation process is one of the mechanism by which axial elongation is thought to cease in embryos [18, 57]. The rate at which the FGF8-to-RA transition occurs, and hence differentiation, is modulated by the strength of mutually repressive FGF-RA interactions (Table 1-IV through VI; Fig 3). Factors that change FGF activity levels (e.g., GDF11; [30, 31]) could effectively changing the strength of repressive interactions between FGF and RA and, therefore, the timing of axial growth termination.

**RA aberrant/oscillatory.**  Some parameters in the FGF-WNT-RA interaction system lead to an oscillation in RA levels that did not match the behavior of the system *in vivo*. These oscillations occurred when Hill constants for RA inputs were weak, particularly for the RA-dependent autoregulation of *Raldh2* production (Table 1-VII, VIII; S3 Fig). In some cases, the system produced a discrete burst of RA at the position where the FGF-RA switch was observed, to then return to produce FGF (Table 1-VII; S3A Fig). In other cases, the burst of RA separated the caudal area of FGF production from a rostral area where FGF and RA production alternated in an oscillatory manner (Table 1-VIII; S3B Fig).

Altogether, our results show that the FGF8-WNT8C-RA interaction network postulated by Olivera-Martinez and colleagues [27] can indeed give rise to a signaling switch that travels caudally during the elongation of the embryonic axis. The model also leaves open the possibility for additional factors to terminate axial elongation (e.g., GDF11; [30, 31]). The behavior of the switch depends on several interaction parameters that, in coordination, regulate the position and size of the region of cell differentiation.

## FGF-WNT-RA signaling switch and transcription factor network establish areas of pluripotency, early and late differentiation

To simulate the dynamics of the transcription factor network, we integrated the transcription (Fig 1C) and signaling (Fig 2A) networks into a single supra-network (Fig 4A). We then used

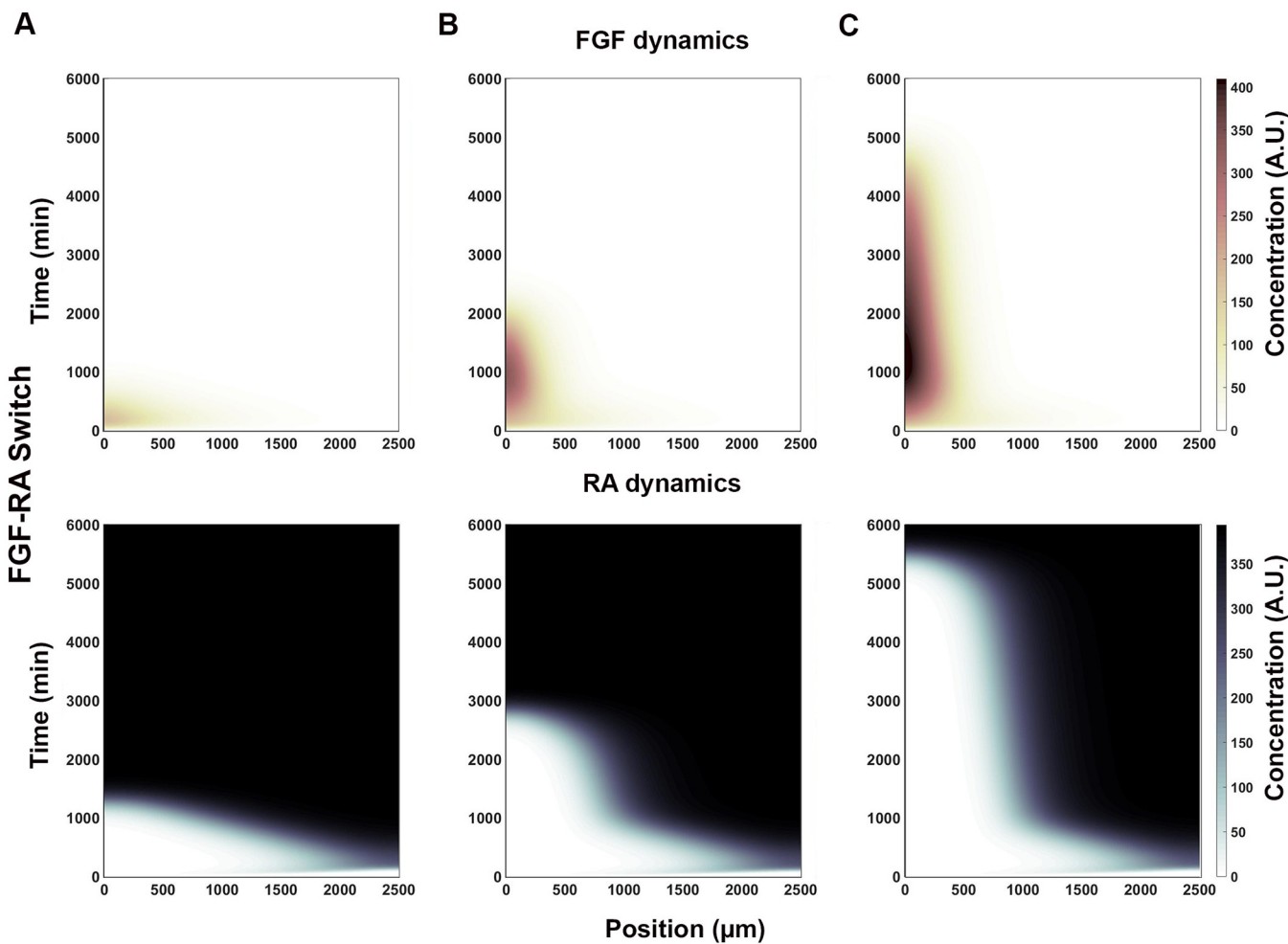

**Fig 3. RA inputs strength determine FGF-RA switch rate of conversion.** The strength by which WNT8C stimulates ($W_{WR}$) and FGF8 represses ($R_{RF}$) *Raldh2* transcription determines RA's spatial profile over time (x and y axes, respectively). (**A**) Fast FGF-RA switch (t<1500 min) results from relatively moderate activation and very strong repression inputs ($W_{WR}$ = 0.5, $R_{RF}$ = 0.1). (**B**) Intermediate FGF-RA switch (t<3000 min) results from relatively strong activation and moderate repression inputs ($W_{WR}$ = 0.2, $R_{RF}$ = 0.3). (**C**) Slow FGF-RA switch (t<5500 min) results from moderate activation and repression inputs ($W_{WR}$ = 0.5, $R_{RF}$ = 0.45). FGF and RA heat map scale is shown on the right (arbitrary units; A. U.).

the FGF-RA balance profile output as the input for the system (Figs 2D and 4B), as it most closely resembles the distribution of signaling activity and NP cell behaviors during the steady state period of embryo growth (10–18 somite stage; Fig 1A). In this simulation, we followed the transcriptional profile of cells as they are born caudally at t = 0 and at subsequent times are displaced rostrally by the appearance of new cells. During their rostral displacement, cells move away from the stem cell zone and the source of FGF and WNT production (Fig 4C). As FGF/WNT level decrease, RA levels increase following the FGF-RA balance profile simulation (Figs 2D and 4B). These changes in spatial signal information are the drivers for transcription factor expression. Since the cells are arranged spatially from caudal to rostral in order of birth, the temporal changes in transcription factors give rise to spatial changes in profiles.

The transcription output of the system depends on signaling inputs and transcription factor interactions. Signals regulate the transcription factor network at two distinct key points. The first point of regulation is towards the caudal end of the chick embryo, where FGF8 and WNT8C, alone or in combination, are required for *T (Bra)*, *Sox2*, *Nkx1.2* and *Cdx4*

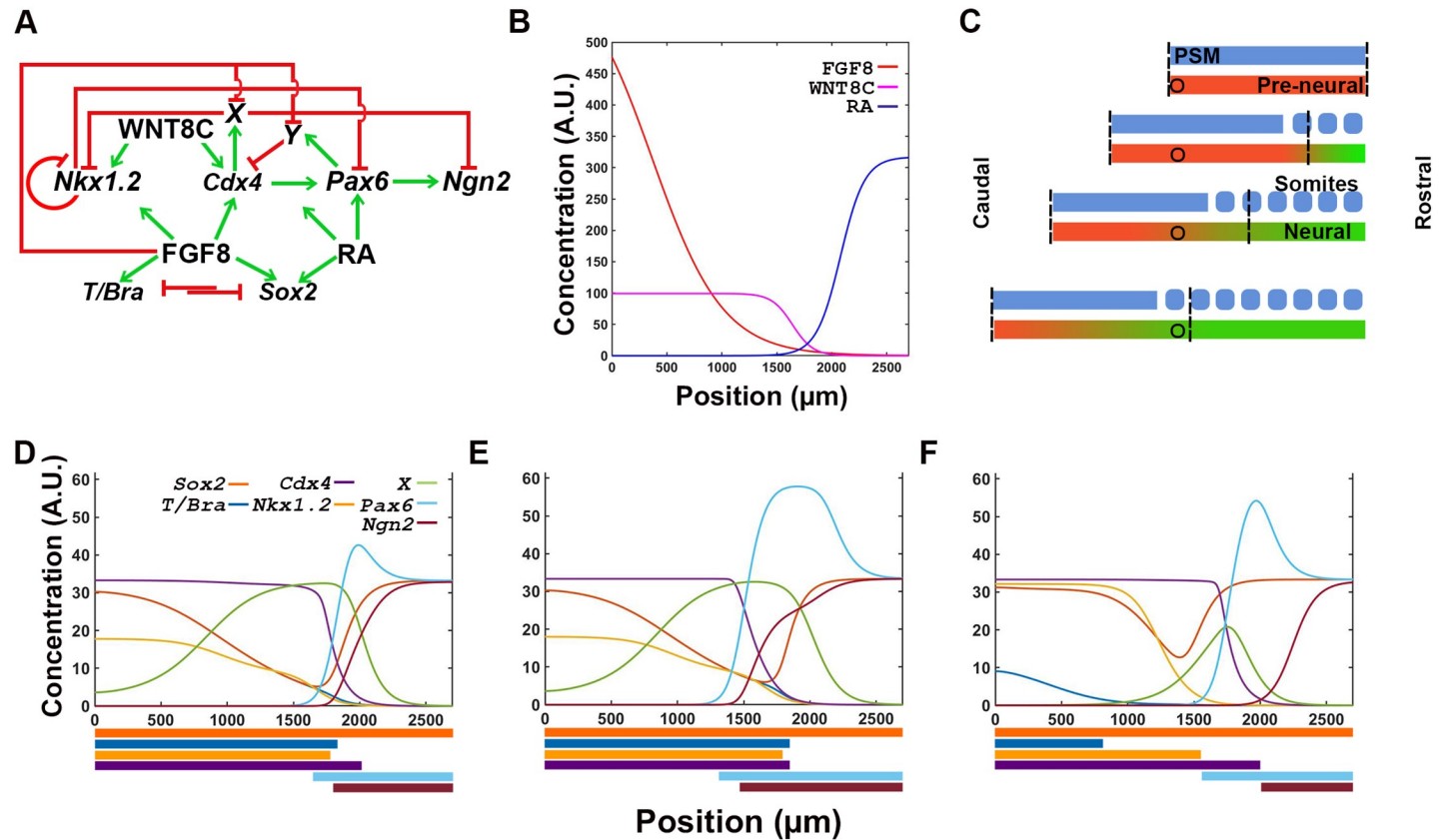

**Fig 4. Strength of interactions between transcription factors determine the response of the network to signal information.** (**A**) Integrated transcription and signaling interaction network, including hypothetical factors X and Y, predicted from experimental data from reference [15]. (**B**) Reference FGF-RA switch input used for simulations (from Fig 2D). (**C**) Location of reference cell (circle) within the spatial maturation domain of signaling activity (vertical dashed lines). Cell remains in the location where it was born as the spatial maturation domain is displaced caudally (as in Fig 2B). (**D-F**) Transcription profile of cell with all interactions equally moderate (D; Hill constants = 20), equally strong (E; Hill constants = 2), and variable (F; as defined in Table 2). Output of simulation in F most closely resembles the staggered distribution of transcription factors observed in embryos (Fig 1B; [15]). Colored bars at the bottom of each graph represent the spatial domain of gene transcription: *Sox2* in dark orange; *T/Bra* in dark blue; *Nkx1.2* in light orange; *Cdx4* in purple; *Pax6* in light blue; and *Ngn2* in maroon.

transcription [17, 58–60]. Independently of signaling inputs, *Nkx1.2* transcription is negatively regulated by CDX4 and its own protein product [15, 61]. The second point of regulation is towards the rostral end of the neural plate, where RA cooperates with CDX4 to activate the early differentiation gene *Pax6* [15, 62]. *Pax6* is also negatively regulated by NKX1.2 [15, 63]. In contrast, transcription of late differentiation gene *Ngn2* is activated by PAX6 and repressed by CDX4 [15, 26]. Given that CDX4 is an activator [64], our model invokes two putative CDX4-regulated transcriptional repressors *X* and *Y* to indirectly repress *Nkx1.2* and *Ngn2* [15]. These hypothetical repressors are assumed to be inhibited by FGF8 [15].

Together, the signal and transcription factor network were able to generate correct gene transcription profile in many but not all instances (Fig 4D–4F), indicating that only under certain parameter restrictions could the network recapitulates embryonic events. In principle, the spatial dynamics of the signal interactions network should be sufficient to activate transcription factor network components in the correct spatiotemporal sequence: high FGF caudally would promote pluripotency while high RA rostrally would promote differentiation, with cross-repressive interaction between pathways maintaining the domains separate at opposite ends of the tissue. However, if all the interactions in the network are equally moderate (Hill constants = 20, Fig 4D) or equally strong (Hill constants = 2, Fig 4E), then the network does

not result in the proper spatial resolution of temporal states. In both cases, transcription of the mesoderm marker *T/Bra* is not restricted to the caudal end, but instead, it is detected throughout the caudal two thirds of the tissue, partially overlapping with *Pax6* and *Ngn2* gene transcripts (Fig 4D, 4E). Only a subset of interaction strengths give rise to correct spatial order of identities (Table 2; Fig 4F). The values of the interactions strengths that generate proper spatial distribution of transcripts could be increased or decreased by 30%. These values define a parametric space where the model is operational and highlights its robustness (S4 Fig). These results suggest that signaling inputs encodes the information required for specifying different cell maturation states, but that it is the transcription factor network what determines the spatial distribution and organization of maturation states cell along the caudal-to-rostral length of the tissue.

## The transcription network executes the spatiotemporal information provided by the signaling factor network

To further evaluate the contribution of signaling and transcription factors networks on cell maturation events, we tested the effect of disrupting individual network nodes on transcription readouts. First, we tested the response of the transcription network to signaling noise. In simulations, both periodic disturbance (Fig 5A) and random noise (Fig 5B) were well tolerated by the transcription network without any distortions in the spatiotemporal resolution of the cellular states. Unexpectedly, introduction of random noise resulted in better separation of early maturation (*Cdx4*$^+$, *Pax6*$^+$, *Ngn2*$^-$) and late differentiation (*Cdx4*$^-$, *Pax6*$^+$, *Ngn2*$^+$) states (Figs 4F, 5B). This phenomenon, the system's ability to withstand perturbations by retaining NP cells in developmental trajectories, suggests that canalization is an emerging property of the signal-transcription factor supra network.

Next, we evaluated the role of signaling gradients in determining the spatiotemporal resolution of downstream targets' transcriptional domains. Replacing the exponential gradient of the signaling factors with a Boolean switch (Fig 5C) or a linear gradients (Fig 5D), resulted in loss of proper resolution of transition zones. Thus, changes in the spatial information contained in the signaling network changes the transcription network readouts. This confirms that the spatial information is encoded in the signal and not in the transcription factor network.

We previously proposed a central role of CDX4 in regulating maturation of NPs in the chick pre-neural tube. To theoretically test CDX4 role in transcription network regulation, we removed, increased or introduced noise to *Cdx4* transcription and evaluated the network's transcription profile output (Fig 6). When *Cdx4* was removed from the simulation, *Nkx1.2* transcription expanded rostrally, overlapping significantly with the expression of differentiation markers *Pax6* and *Ngn2* (Fig 6A). This phenomenon is opposite to what is observed experimentally, were downregulation of CDX4 activity using an ENRCDX4 repression construct results in downregulation of *Nkx1.2* [15] (discussed below). Conversely, when the levels of *Cdx4* were increased in the simulation, *Nkx1.2* expression domain shifted caudally and away from *Pax6* expression domain, and rostral cells did not activate the late differentiation gene *Ngn2* (Fig 6B), in agreement with experimental results [15]. Thus, removing or increasing *Cdx4* transcription affects the spatial relationship between early specification gene *Nkx1.2* and neural differentiation gene *Ngn2*. This result suggests that CDX4 functions in the network to establish a transition zone between pluripotency and differentiation states. CDX4 function is robust and integral to the canalization properties of the system, as introduction of transcriptional noise produces the expected gene expression profile with only minute changes in the position of boundary transitions (<+/-30μm; Fig 6C). Excluding the effect of removing CDX4 on *Nkx1.2* (discussed below), our simulations agrees with *in vivo* observations [15], and support a role of CDX4 in driving NP maturation during early spinal cord development.

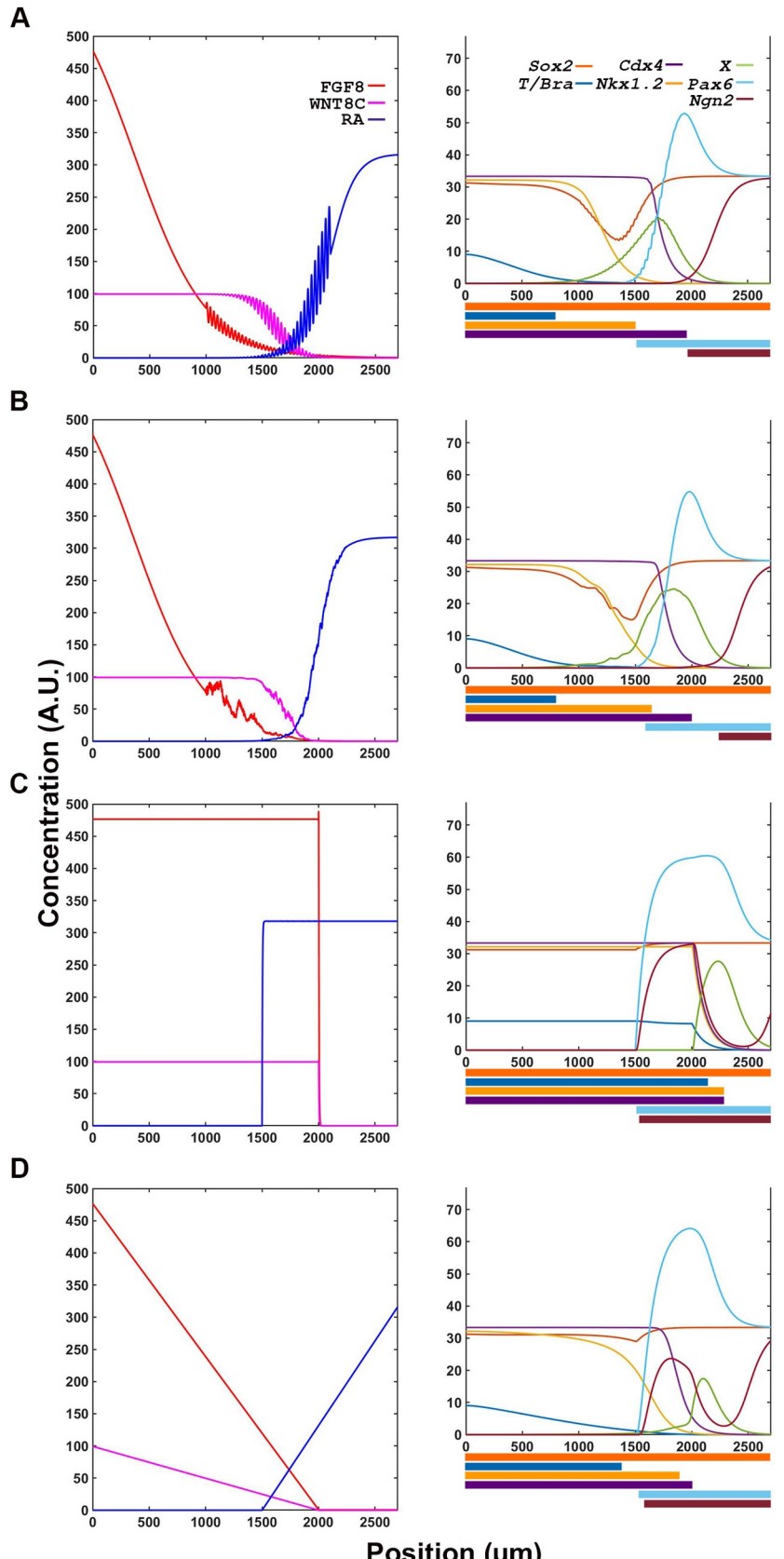

**Fig 5. Transcription factor network is resilient to small and moderate alterations in signaling information.** (**A, B**) Transcription factor output is not affected by oscillatory (A) or random noise (B) in signaling inputs, as outputs are comparable to those obtained in conditions without noise (Fig 4F). (**C, D**) Large changes in signaling input such as discreet Boolean (C) and linear gradient (D) changes transcription factor expression domains.

## Discussion

### Signaling factor simulation recapitulates signaling dynamics observed in natural systems

Our simulations describe the possible behaviors the FGF8-WNT8C-RA system can exhibit under various interaction conditions (Fig 2). With small variations in interactions' strength, the system can model behaviors associated with different stages of axial tissue development. In a system where the FGF's activity dominates over RA's activity, the simulation most closely resembles the neural tissue at early stages of axial extension (in chick, before 6 somite stages), whereas in a system where FGF activity balances that of RA, the simulation resembles the maturation of spinal cord cell that occurs during axial elongation before the formation of the tail bud (in chick, 6–18 somite stages). In contrast, a switch in the system from FGF to RA most closely resembles the processes occurring during termination of body axis extension [10, 28], with or without the aid of additional factors (e.g., GDF11; [30, 31]). Significantly, when the interactions between FGF and RA components are weak, the system oscillates, resembling the oscillations observed between FGF/WNT and the NOTCH signaling pathway during the process of paraxial mesoderm segmentation [65]. Thus, with small modifications in signal components interaction, one can observe large changes in the behavior of the system equivalent to the changes normally observed in the tissues emerging from the caudal lateral epiblast during axial elongation, the paraxial mesoderm and spinal cord.

We propose a model of vertebrate body extension where modulation of interaction strength between different components of the system (e.g., transcriptionally, post-transcriptionally or epigenetically), could regulate the spatiotemporal dynamics involved in vertebrate body extension. In this model, the time at which the system transitions from FGF dominant, to FGF-RA balance, to RA switch respectively determine the time of tissue induction, elongation and termination. For example, a long period in which the FGF8-RA balance system is operational could explain the elongated axis of vertebrates such as snakes; as long as the FGF8-RA balance system remains operational, the caudal progenitor/stem cell pool will continue to generate tissue and extend the axis. In this scenario, the time at which RA takes over the system to initiate progenitor cell differentiation will determine the axial body length. This last process can be accelerated by other factors that dampen FGF and Wnt signaling activity such as GDF11 [30, 31]. A second mechanism for terminating axial elongation is the activation of *Hox13* genes [66], whose activation in mouse is under the control of CDX factors as well as GDF11 [31, 66].

### Transcription network simulations recapitulate the cell state transitions observed in the caudal neural plate

Results from simulations support a role for CDX in coordinating upstream signaling factors with downstream transcription network components involved in spinal cord neural maturation. In the present model, CDX4 functions to separate caudal stem cell populations (*Nkx1.2*$^+$ *Pax6*$^-$ *Ngn2*$^-$) from rostral differentiating cells (*Nkx1.2*$^-$ *Pax6*$^+$ *Ngn2*$^+$) by establishing a transition zone. This is achieved by CDX4 repressing the bipotency gene *Nkx1.2* and the late differentiation gene *Ngn2*, and by activating the early differentiation gene *Pax6*. In simulations, high levels of *Cdx4* transcription resulted in downregulation of CDX4 repressed genes (Fig 6B):

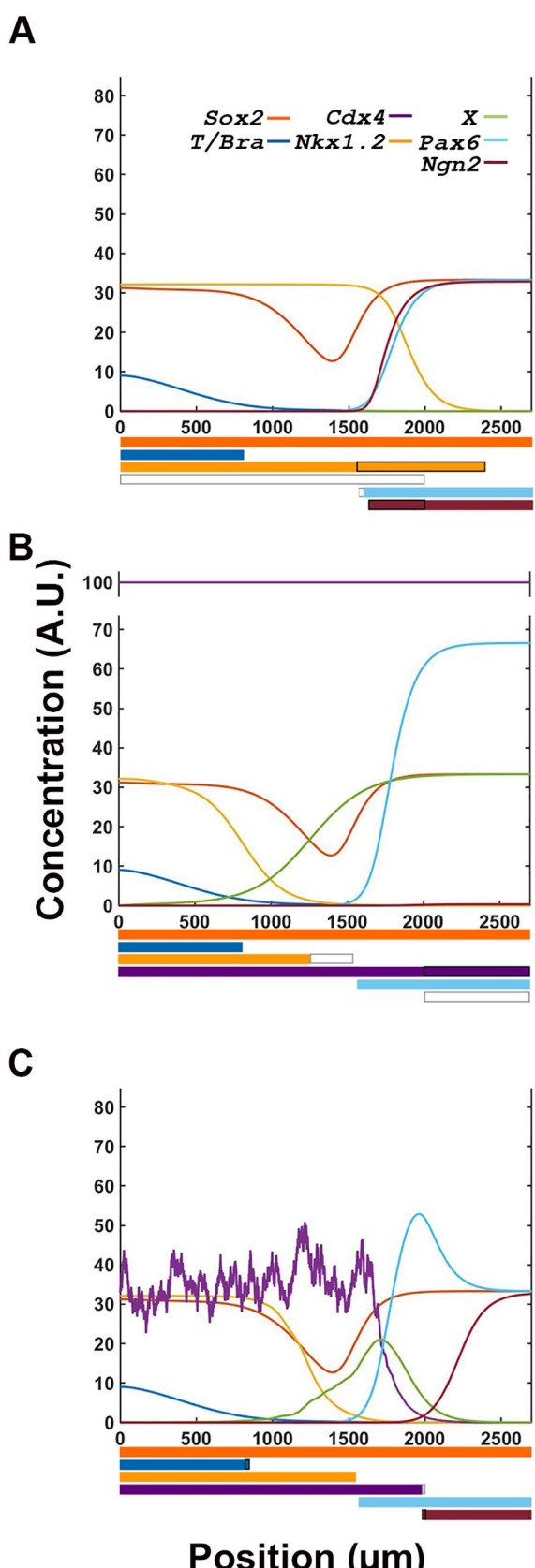

**Fig 6. CDX4 is necessary for proper interpretation of signaling inputs by the transcription factor network.** (**A**) Compared to control simulation (Fig 4F), loss of *Cdx4* expression causes a large rostral expansion of *Nkx1.2* domain, a small reduction in *Pax6* domain and a large caudal expansion in *Ngn2* domain. These changes results in the overlap of stem and differentiation gene expression domains. (**B**) Overexpression of *Cdx4* reduces *Nkx1.2* and eliminates *Ngn2* expression domains, effectively expanding the early and eliminating the late differentiation zones. (**C**) Introduction of random noise in *Cdx4* transcriptional noise has insignificant effects on the system's spatial expression profiles. In the expression profile bars at the bottom of the graphs, white and black rectangles indicate loss and gain of gene transcription, respectively.

*Nkx1.2* transcription domain shifted caudally and *Ngn2* transcription was lost. In these conditions, only the caudal expression of *Nkx1.2* was retained due to its high dependence on WNT stimulation [61]. Increasing *Cdx4* transcription did not affect the expression domain of *Pax6*, as transcription of this gene is also dependent on RA secreted from somites [15, 62]. Together, the changes in *Nkx1.2* and *Ngn2* transcription induced by CDX4 overexpression effectively increase the size of the transition zone. The same way that premature activation of differentiation signals has been predicted to cause shortening of the embryonic axis [18], a greater separation of stem cell and differentiation signals is predicted to cause axial lengthening. These predictions would need to be tested experimentally.

Significantly, results obtained by simulating loss of *Cdx4* activity (Fig 6A) did not fully match experiments done *in vivo*. With respect to differentiation genes, the network recapitulates the *in vivo* results: *Pax6* transcription was not affected due to dependence of this gene on RA [15, 62], whereas *Ngn2* transcription was upregulated because this gene is normally repressed by CDX4 [15]. In contrast, with respect to the NMP marker *Nkx1.2*, loss of *Cdx4* caused an anterior expansion of *Nkx1.2* expression domain that was not observed experimentally. This discrepancy can be attributed to the use of a dominant negative form of CDX4 instead to knockout allele to downregulate the activity of this gene *in vivo* (ENRCDX4; [15]). Dominant-negative ENRCDX4 works by outcompeting endogenous CDX4 from binding to its target genes and repressing their transcription [67]. This approach is different than not having CDX4 protein altogether (e. g, through deletion of the gene). Given that our model simulates the loss of CDX4 function and not the active repression of its downstream target genes, this providing a possible explanation for the observed discrepancies between experimental systems. It is also possible that our current understanding of the transcription factor network is incomplete. For example, NOTCH signaling pathway is involved in NP cell proliferation [32, 33], but was omitted from our system due to lack of information related to its interaction with *Cdx4* and *Nkx1.2*. It is possible, however, that NOTCH is a positive regulator of *Nkx1.2* in a manner similar to its regulation of *Nkx6.1*, a close family member involved in ventral spinal cord cell specification [68]. These two possible explanations are not mutually exclusive, and could be resolved with additional experiments. While additional experiments will be required to fully understand CDX4 function in the NMP zone, even with its limitations, the proposed transcription factor network supports a key role for CDX4 in the segregation of cell states in the nascent spinal cord.

Noise is an intrinsic property of biological systems [69]. To explain the resilience of developmental systems to genetic or environmental noise and perturbations, Waddington introduced the concept of canalization [8]. In our simulation, the introduction of random noise produced a more accurate representation of the cell maturation states observed *in vivo* than those produced without any type of noise (e.g., separation of late maturation and differentiation states; Figs 4F, 5B). In addition, deviation of up to 30% in the system's parametric values (Table 2) did not change the spatial distribution of cell maturation states. This exceptional robustness was an unexpected emerging property of the system that was not obvious from experimental data [15]. We propose that the network's resiliency to intrinsic (random noise)

and extrinsic (artificial variation in parameters) variations could function to canalize NPs to their mature state, and that the source of the system's canalization capacity is the network's organization itself [70].

## Future perspectives

The findings uncovered in our chick embryo model are generalizable to other vertebrates, despite embryonic differences in tissue size, geometry and heterochrony. In all species examined so far, similar but not identical signal and transcription factors control spinal cord cell specification and maturation [9, 10, 25, 71]. Similar to chick, mouse and zebrafish NMP bipotent state is driven by FGF/WNT pathways regulating *T/Bra* and *Sox2* transcription, with differences residing in the specific FGF or WNT regulating each pathway [19–21, 71, 72]. For example, WNT8c in chick and WNT3a in mouse control axial elongation and NMP cell fate decisions (reviewed in [10]). Different WNT proteins are post-translationally modified in a number of ways, and these variations can change their extracellular transport and diffusion (reviewed in [73]), which would directly affect the shape of their gradient. While this idea need to be tested experimentally, it is possible that differences in individual network components help adapting an otherwise conserved network to tissues with different morphologies and rates of development.

Although the integrated signaling and transcription factor network model presented here provides key information on the transition state drivers underlying neuronal cell maturation, it is clear from experimental and modelling data that the model is far from complete. For example, several signaling and transcription factors were omitted from the system due to either lack of information regarding their interactions with other network members (e.g., NOTCH; [32, 33]), or reports that those factors are not operational during the developmental stages that the system analyses (chick 10–18 somite stage; e.g., other CDX family members, [29]; GDF11, [30, 31]; Hox13 genes, [31, 66]). Another missing component are the feedback controls that transcription factors have over the signaling network. In mouse, chromatin immunoprecipitation studies using epiblast stem cells derived from wild type or CDX2-deficient primitive streaks have shown that CDX transcription factors can regulate several WNT and FGF pathway components (*Wnt5a*, *Rspo3*, *Fgf4* and *Fgf8*; [74]), indicating feedback regulation between transcription and signaling networks. Similarly, CDX binding sites present in *Radh2* intronic enhancer are sufficient to drive reporter gene expression in the caudal end of embryos [75]. Currently, however, lack of quantitative data precluded the incorporation of feedback activities into an integrated network model.

Our modeling results also highlights the importance of signaling factor regulation by components external to the signaling pathways. For example, our model shows that maintenance of RA production is critical for the behavior of the system, as weakening of the Hill constant regulating RA-dependent autoregulation of *Raldh2* production causes the system to transition from balanced to oscillatory (Table 1-VII, 1-VIII; S3 Fig). While, for simplification purposes we assumed that Raldh2 maintenance is dependent on RA, it is likely to be dependent on transcription factors, some of which are part of our transcription network (e.g. CDX; [75]). Understanding the effect that transcription factor network components have over the signaling network will be important for understanding the later stages in neural cell maturation and their subsequent differentiation.

## Supporting information

**S1 File.**
(DOCX)

**S1 Fig. Hill constant determine the strength of response of targets to activators and repressors.** Temporal response of targets with different Hill constants to activators and repressors. Inputs are shown in blue and targets with different Hill constants are color coded: H1 = 1, orange; H2 = 10, yellow; H3 = 20, purple; and H4 = 100, green. **(A-B)** For activators, constant (A) and graded (B) inputs induce targets with smaller Hill constants to higher levels than targets with larger Hill constants. (**C-D**) For repressors, constant (C) and graded (D) inputs reduce targets with smaller Hill constants to lower levels than targets with larger Hill constants. With graded inputs (B, D), larger Hill constants also cause temporal delays in response.
(TIF)

**S2 Fig. Changes in FGF-WNT-RA signaling interactions results in mRNA profiles parallel protein accumulation and are stable over time.** (**A**) Profiles of mRNA transcripts at t = 6000 min associated with production of signaling molecules. Transcript and protein profiles are similar (Fig 2C left panels). (**B**) Signaling molecule profiles are stable over longer simulation times. An FGF-RA balance simulation that was run for t = 30,000 min produced the same profile than a simulation that was run for t = 6000 min (Fig 2C middle row).
(TIF)

**S3 Fig. RA positive autoregulation is required for bistability.** Reducing RA's positive effect on *Raldh2* transcription (H = 300 instead of 50) results in aberrant RA, but not FGF, distribution. (**A**) Under these conditions, when FGF affinity to repress *Raldh2* is strong (H = 2 instead of 10), a peak of RA production forms at a position in the field where the FGF-RA switch would have occurred (1500–2000 μm). (**B**) When RA repression of *Fgf8* transcription is weakened (H = 20 instead of 1), RA production oscillates in the region of cell differentiation (>1500 μm).
(TIF)

**S4 Fig. Stability of the transcription profile in the parameter space.** Changes in the strength of interactions between transcriptional factors does not drastically affect the transcriptional domain profile. (**A**) Original transcription profile as shown in Fig 4F. (**B, C**) Reducing (B) or increasing (C) all the Hill constants in the interaction network by 30% does not significantly change the spatial profile of gene transcription.
(TIF)

**S1 Appendix. SIGNET.m: MATLAB code for simulating signaling dynamics.**
(M)

**S2 Appendix. TRANSNET.m: MATLAB code for simulating transcriptional factor dynamics.**
(M)

## Acknowledgments

We thank Dr. Donald DeAngelis for guidance and for sharing his mathematical expertise, and members of the Skromne lab for intellectual discussion. We also thank Dr. K. G. Story (U Dundee, UK), Dr. M. Gouldin (Salk Institute, USA). Dr. F. Medeville (CBI, France), Dr. S. Mackem (NCI, USA), Dr. Y. Marikawa (U Hawaii, USA), Dr. A. V. Morales (Cajal Institute, Spain) and Dr. B. Novitch (UCLA, USA) for generously providing plasmids.

## Author Contributions

**Conceptualization:** Piyush Joshi, Isaac Skromne.

**Data curation:** Piyush Joshi.

**Formal analysis:** Piyush Joshi, Isaac Skromne.

**Funding acquisition:** Isaac Skromne.

**Investigation:** Piyush Joshi, Isaac Skromne.

**Methodology:** Piyush Joshi.

**Software:** Piyush Joshi.

**Supervision:** Isaac Skromne.

**Validation:** Piyush Joshi, Isaac Skromne.

**Visualization:** Piyush Joshi.

**Writing – original draft:** Piyush Joshi.

**Writing – review & editing:** Piyush Joshi, Isaac Skromne.

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
