## [Decision Letter · Decision Letter 0]

28 Aug 2020

PONE-D-20-22620

A theoretical model of neural maturation in the developing spinal cord

PLOS ONE

Dear Dr. Skromne,

Thank you for submitting your manuscript to PLOS ONE. After careful consideration, we feel that it has merit but does not fully meet PLOS ONE’s publication criteria as it currently stands. Therefore, we invite you to submit a revised version of the manuscript that addresses the points raised during the review process.

This manuscript offers a theoretical model for spinal cord development, based on complex mathematical modelling of molecular signaling. The work is well written and of high quality and offers a novel and substantial contribution to the scientific field that future works may build upon. However, several important considerations were raised by the reviewers, which should (at least partiallly) be addressed by the authors. In particular, the mathematical model should incorporate all important molecular signals and/or justify why they are excluded. Furthermore, this paper needs to discuss the competing theory of the derivation of spinal neural progenitor cells of "gradual patterning and canalization" and address why this was not utilized. Finally, the authors need to demonstrate the theoretical framework in an vivo model or rationalize why this cannot be performed (as a limitation).

We look forward to receiving your revised manuscript.

Kind regards,

Allan R. Martin

Academic Editor

PLOS ONE

Journal Requirements:

2.Thank you for including the following ethics statement on the submission details page:

'The American Association for Laboratory Animal Sciences (AALAS) and the American

Veterinary Medical Association (AVMA) do not consider 24-hour old chick embryos,

such as the ones used in this study, "vertebrates" capable of feeling pain. Therefore,

embryos in this study are exempt of Institutional Animal Care and Use Committees

(IACUC) review.'

Please include this information in the ethics statement in the Methods section of your manuscript and also specify the source of chick embryos used for your study.

3.Thank you for stating the following in the Funding Section of your manuscript:

[P. J. was supported by Sigma XI GIAR. I. S. was supported by University of Richmond School

of Arts and Sciences and by the National Science Foundation (IOS-1755386).]

 [The funders had no role in study design, data collection and analysis, decision to publish, or preparation of the manuscript.]

Reviewers' comments:

Reviewer's Responses to Questions

**Comments to the Author**

1. Is the manuscript technically sound, and do the data support the conclusions?

Reviewer #1: Partly

Reviewer #2: Partly

2. Has the statistical analysis been performed appropriately and rigorously? 

Reviewer #1: Yes

Reviewer #2: Yes

3. Have the authors made all data underlying the findings in their manuscript fully available?

Reviewer #1: Yes

Reviewer #2: Yes

4. Is the manuscript presented in an intelligible fashion and written in standard English?

Reviewer #1: Yes

Reviewer #2: Yes

5. Review Comments to the Author

Reviewer #1: Using spatiotemporal dynamics of FGF/WNT/RA morphogen signalling and CDX transcription factor (TF) networks, the authors proposed a theoretical model to determine how these signalling gradients could be used to antagonize, or create synergistic effects, and the result on cell differentiation and potency. These effects were calculated subsequently, as a partial differential equation (used to later determine the result of functions that involve several variables), was used to determine the output of morphogen signalling, where this output was then used for the input of an ordinary differential equation to determine the resulting effects of the gradients on TF networks. Notably, the authors have identified broad behaviours of morphogen signalling, involving FGF-dominance, FGF/RA balance, switch from FGF to RA, and aberrant/oscillatory RA signalling, which describes a transition from pluripotency, homeostasis of morphogens, differentiation, and repression of differentiation activation, respectively. The resulting effects on transcription factor networks is a recapitulation of what is described in the literature, highlighting the importance of CDX4 to produce caudal stem cell populations (e.g. Nkx1.2+/Pax6-/Ngn2-).

Ultimately, this paper provides important first steps in the mathematical modelling of developmental ques in the spinal cord, with the limited quantitative knowledge available in literature to accurately and precisely model development. To optimize the authors’ dissemination of information in this publication, the below considerations should be addressed to proceed with publication:

• This paper is mainly based on the theory of derivation of spinal neural progenitors from “Neruomesodermal progenitors” (NMP) that is first introduced by Mina Gouti and is now continued by her lab. Although there is robust experimental evidence for this model, another model for derivation of spinal neural progenitor cells is that of gradual patterning and canalization. The authors need to explain these two models and discuss how their mathematical model fits to any or both of those experimental models.

• For someone who is unfamiliar with modelling biological processes with differential equations, it may be difficult to initially understand why a combination of partial differential equations are necessary to input into an ordinary differential

o A brief background regarding the subsequent role of partial and ordinary equations can help disseminate your findings to the lay population of biologists.

• The modeling is heavily based on the expression level of “Cdx4” that the authors found contributed in the their last experimental paper (Joshi et al. 2019). It seems that in the current paper the authors want to justify the importance of Cdx4, but ignore other important transcription factors like Hox genes, and Cdx1.

• The assumptions used for the differentiation of spinal neural progenitor cells in this paper are over-simplified. The purpose of mathematical modeling is to predict complex biological/developmental events. Oversimplifying the assumptions and ignoring some details and pathways in favor of other factors is contradictory to the purpose of mathematical modeling.

• The manuscript does not consider the BMP morphogen gradients of growth differentiation factor 11 (GDF11) in the AP-axis nor BMP DV-axis developmental modelling. Also, the effect of Notch signaling in patterning is heavily ignored. TGF-b gradient, DKK1, FRZB, GDF7 and … factors should be considered in the equation.

• To simplify their model, the authors consider the “most influential” factors FGF8/WNT8C to control FGF/WNT signalling in development. Depending on the species, different factors, such as FGF17, FGF18, FGF4 or WNT3A contribute to the morphogen regulation. These factors should also be considered when making a reliable predictive modeling.

• The authors do not specify the species that they are matching their signalling gradients to in their equations.

o The authors should mention, more clearly, which species you are trying to model developmentally. The majority of their references seem to refer to either the chick or mouse, but different experimental evidence exists for either species, which may contradict, or have missing information, based on species variation.

• Simplification of morphogen gradients that are only regulated by other protein-protein interaction dynamics. The paper fails to consider in their mathematical modelling how different neural cell fate transcription factors may affect these signalling ques during development.

• Confusing terminology regarding Hill constant. Is this the output of the Hill coefficient when all theoretical values are included in the equation?

Reviewer #2: The authors provide a mathematical model of spinal cord maturation and the role of signaling and interaction between transcription factors. Hill constants were used to determine the strength of interaction between the compounds.

The authors highlight the models used in great detail, which may be overwhelming to some readers.

The biggest drawback of the paper is its external validity that should be tested in an in vivo model.

6. PLOS authors have the option to publish the peer review history of their article (what does this mean?). If published, this will include your full peer review and any attached files.

Reviewer #1: No

Reviewer #2: No

---

## [Author Response · Author response to Decision Letter 0]

27 Oct 2020

Reviewers responses can be found in the document "Response to Reviewers"

---

## [Decision Letter · Decision Letter 1]

7 Dec 2020

A theoretical model of neural maturation in the developing chick spinal cord

PONE-D-20-22620R1

Dear Dr. Skromne,

We’re pleased to inform you that your manuscript has been judged scientifically suitable for publication and will be formally accepted for publication once it meets all outstanding technical requirements.

Kind regards,

Allan R. Martin

Academic Editor

PLOS ONE

Additional Editor Comments (optional):

Reviewers' comments:

Reviewer's Responses to Questions

**Comments to the Author**

1. If the authors have adequately addressed your comments raised in a previous round of review and you feel that this manuscript is now acceptable for publication, you may indicate that here to bypass the “Comments to the Author” section, enter your conflict of interest statement in the “Confidential to Editor” section, and submit your "Accept" recommendation.

Reviewer #1: All comments have been addressed

2. Is the manuscript technically sound, and do the data support the conclusions?

Reviewer #1: Yes

3. Has the statistical analysis been performed appropriately and rigorously? 

Reviewer #1: Yes

4. Have the authors made all data underlying the findings in their manuscript fully available?

Reviewer #1: Yes

5. Is the manuscript presented in an intelligible fashion and written in standard English?

Reviewer #1: Yes

6. Review Comments to the Author

Reviewer #1: (No Response)

7. PLOS authors have the option to publish the peer review history of their article (what does this mean?). If published, this will include your full peer review and any attached files.

Reviewer #1: No

---

## [Editor Report · Acceptance letter]

10 Dec 2020

PONE-D-20-22620R1 

A theoretical model of neural maturation in the developing chick spinal cord 

Dear Dr. Skromne:

I'm pleased to inform you that your manuscript has been deemed suitable for publication in PLOS ONE. Congratulations! Your manuscript is now with our production department. 

Kind regards, 

on behalf of

Dr. Allan R. Martin 

Academic Editor

PLOS ONE